# Post-Harvest Application of Nanoparticles of Titanium Dioxide (NPs-TiO_2_) and Ethylene to Improve the Coloration of Detached Apple Fruit

**DOI:** 10.3390/foods12163137

**Published:** 2023-08-21

**Authors:** Yongxu Wang, Guolin Chen, Daru Wang, Jing Zhang, Chunxiang You, Xiaofei Wang, Huaifeng Liu

**Affiliations:** 1Xinjiang Production and Construction Corps Key Laboratory of Special Fruits and Vegetables Cultivation Physiology and Germplasm Resources Utilization, Department of Horticulture, College of Agriculture, Shihezi University, Shihezi 832003, China; 15615541161@163.com; 2Apple Technology Innovation Center of Shandong Province, Shandong Collaborative Innovation Center of Fruit & Vegetable Quality and Efficient Production, National Key Laboratory of Wheat Improvement, College of Horticulture Science and Engineering, Shandong Agricultural University, Tai’an 271018, China15953327292@163.com (D.W.); zjlucky815@163.com (J.Z.); youchunxiang@126.com (C.Y.); xfwang2004@163.com (X.W.)

**Keywords:** apple coloration, nanoparticle, NPs-TiO_2_, reactive oxygen species, ethylene

## Abstract

In this study, we analyzed the effects of treatments with titanium dioxide nanoparticles (NPs-TiO_2_) and ethylene on anthocyanin biosynthesis and reactive oxygen species (ROS) metabolism during light exposure in ripe ‘red delicious’ apples. Both treatments led to improved anthocyanins biosynthesis in detached mature apples, while the NPs-TiO_2_ had less impact on the fruit firmness, TSS, TA, and TSS/TA ratio. Furthermore, the effects of both treatments on the expression of anthocyanin-related enzymes and transcription factors in the apple peel were evaluated at the gene level. The differentially expressed genes induced by the two treatments were highly enriched in the photosynthesis and flavonoid biosynthesis pathways. The expression of structural genes involved in anthocyanin biosynthesis and ethylene biosynthesis was more significantly upregulated in the ethylene treatment group than in the NPs-TiO_2_ treatment group, and the opposite pattern was observed for the expression of genes encoding transcription factors involved in plant photomorphogenesis pathways. In addition, the ROS levels and antioxidant capacity were higher and the membrane lipid peroxidation level was lower in fruit in the NPs-TiO_2_ treatment group than in the ethylene treatment group. The results of this study reveal differences in the coloration mechanisms induced by NPs-TiO_2_ and ethylene in apples, providing new insights into improving the color and quality of fruits.

## 1. Introduction

The apple is an economically important fruit crop that is widely consumed. According to the FAO, the worldwide apple production quantity is about 82 million tons, of which China produces more than 42 million tons [1,2]. Consumers usually associate the red color of apple fruit with ripeness and a sweet taste, thereby establishing red apple cultivars as particularly favored among consumers [3]. The biosynthesis of anthocyanins in the apple peel is affected by a variety of environmental factors and plant hormones [4], with solar illumination being one of the most important environmental factors. Reflective mulch is usually used to promote apple fruit coloration and improve the utilization effect of light [5,6]. In addition, ultraviolet (UV) radiation and low temperatures are particularly important for regulating the production of anthocyanins in apples [7,8].

The typical approaches employed to artificially improve the deposition of pigments in the apple epidermis involve the use of cultivation techniques that facilitate efficient light utilization to promote anthocyanin biosynthesis, as well as the application of chemically synthesized biostimulants or the like to promote ripening and accelerate anthocyanin accumulation [9]. A variety of cultivation strategies have been used to promote the deposition of pigments in apple fruits, such as approaches that increase the transmissibility of light [5,10], micro-sprinkler irrigation systems [11], fruit bagging [12], delayed harvesting [13], and spray applications of chemicals such as exogenous ethylene [14,15]. However, spray applications of chemically synthesized materials promote defoliation, which improves the penetration of sunlight and accelerates the ripening of fruits, which reduces their storage tolerance [16,17]. In addition, consumers might be discouraged to purchase fruits that have been treated with ethylene-releasing compounds because of their potential health hazards [18].

Nanotechnology is expected to provide important alternatives to conventional cultivation methods [19]. Engineered nanomaterials have been widely used in agriculture for improved crop quality and efficiency and to enhance the ability of crops to deal with environmental stress [20,21]. Titanium dioxide nanoparticles (NPs-TiO_2_) are some of the most widely used engineering nanomaterials [22]. NPs-TiO_2_ are chemically stable, non-toxic, and highly reactive, and have been shown to effectively absorb UV radiation [23]. When NPs-TiO_2_ are exposed to UV light with a wavelength equal to or less than 387.5 nm, reactive oxygen species (ROS) can be generated via the photocatalysis of H_2_O [24,25]. These photocatalytically generated ROS can attack and degrade organic material (e.g., plant organs, microorganisms, viruses, and organic compounds) adsorbed on the surfaces of NPs-TiO_2_ in both aqueous and gaseous phases [26,27,28,29]. A previous study has shown that the attachment of NPs-TiO_2_ to the surface of the exocarp can accelerate the coloring of the exocarp by absorbing UV light, and the NPs-TiO_2_ particles attached to the fruit’s epidermal surface can be washed off using a common tap process [30]. In addition, Surround™ WP (95% kaolin) was applied to reduce sunburn on fruit by reflecting more light [31]. The mechanisms by which NPs-TiO_2_ treatment regulate anthocyanin metabolism in apple fruit during light exposure remain unclear. Here, differences in the coloring mechanisms of apples and the regulation of anthocyanin biosynthesis under treatments with NPs-TiO_2_ and the conventional plant growth regulator ethylene were investigated. The results subsequently revealed differences in the coloring mechanisms of apples induced by NPs-TiO_2_ and ethylene, which have implications for the development of new agricultural applications of photocatalytic nanoparticles.

## 2. Results

### 2.1. Changes in Peel Color and Fruit Quality during Light Exposure

Figure 1 shows the changes in apple fruit phenotype and skin color during light exposure. The rind coloring process was significantly accelerated in both treatments compared with the control (Figure 1A). The color parameters (Figure 1B–E) measured included the brightness (*L**), redness (*a**), yellowness (*b**), and hue angle (*h**). During light exposure, the *a** value was highest in the ethylene treatment, followed by the NPs-TiO_2_ treatment and the control (Figure 1C). The *h** value of the control decreased more slowly than that of the NPs-TiO_2_ and ethylene treatments, and there was a significant difference (*p* < 0.05) on the second day of treatment (Figure 1E). The results indicated that the utilization of NPs-TiO_2_ and ethylene treatments led to noticeable changes in the color and quality of the fruit epidermis during light exposure.

Fruit quality at the end of the 8-day treatment period was measured. The firmness of fruit was significantly lower in the ethylene treatment (*p* < 0.05) than in the NPs-TiO_2_ treatment and control (Figure 1F). The total soluble solids (TSS) (Figure 1G) and solid–acid ratio (Figure 1I) were significantly (*p* < 0.05) higher in the ethylene treatment than in the NPs-TiO_2_ treatment and the control; however, there were no significant differences (*p* ≥ 0.05) between the NPs-TiO_2_ treatment and the control. In addition, the fruit respiration rate (Appendix A) and ethylene production (Appendix A) were significantly higher (*p* < 0.05) in the ethylene treatment than in the other two treatments at 8 days. These findings indicate that the fruits in the ethylene treatment were more mature after they were harvested at the end of the in vitro light culture.

### 2.2. Effects of NPs-TiO_2_ and Ethylene Treatments on the Secondary Metabolites in Exocarp Tissue

The color of the skin is closely related to pigment deposition, and the most common pigments in the skin of apple fruits are anthocyanins. Flavonoids and phenolic compounds can also affect the coloration of apple fruits [32]. The content of anthocyanins, proanthocyanins, flavonoids, and total phenols was determined during light exposure (Figure 2). The anthocyanin and proanthocyanin content of the exocarp treated with the NPs-TiO_2_ and ethylene treatments significantly differed (*p* < 0.05) from that in the control on the second day of treatment (Figure 2A,B). After 8 days of light exposure, the anthocyanin content of fruits sprayed with ethephon reached 293.44 ± 18.01 μg g^−1^, which was 1.11 times and 1.46 times higher than that in the NPs-TiO_2_ treatment and the control, respectively (Figure 2A). Variation in the content of flavonoids and anthocyanins among treatments was similar, and the content of flavonoids and anthocyanins was significantly higher in the ethylene treatment (*p* < 0.05) than in the NPs-TiO_2_ treatment and the control on the sixth day (Figure 2C). The content of total phenolics in the fruit epidermis continued to increase over time, and the content measured in the NPs-TiO_2_ treatment group was significantly higher (*p* < 0.05) than that of the remaining two treatments on the sixth day. On the eighth day of treatment, the total phenolics content of fruit in NPs-TiO_2_ group finally reached 1661.17 ± 77.87 μg g^−1^, which was 1.10 times and 1.24 times of ethylene and control, respectively (Figure 2D).

### 2.3. Transcriptome Analysis of the Differently Expressed Genes (DEGs)

To identify DEGs between treatment groups, transcriptomic analyses were conducted using samples collected during 8 days of treatment. Correlation analysis of the data in each group, principal component analysis scores, and a Venn diagram of DEGs between treatments revealed that the transcriptome data were reliable (Figure 3A,B). Venn diagram showed that 1124 DEGs were overlapped between clusters of control vs. NPs-TiO_2_ and control vs. ethylene, which was much higher than the number of DEGs compared between the other two groups (Figure 3C). A volcano plot was constructed to clarify the distribution of DEGs among the control, NPs-TiO_2_ treatment, and ethylene treatment (log_2_(Fold Change) > 1, *q*-value < 0.05). The results revealed 3416 DEGs between the control and NPs-TiO_2_ treatment, including 2298 up-regulated DEGs and 1118 down-regulated DEGs (Figure 3D). A total of 1719 DEGs were detected between the control and ethylene treatment, including 1264 up-regulated DEGs and 455 down-regulated DEGs (Figure 3E). A total of 934 genes were detected between the NPs-TiO_2_ and ethylene treatments, including 379 up-regulated DEGs and 555 down-regulated DEGs (Figure 3F).

### 2.4. Gene Ontology (GO) and Kyoto Encyclopedia of Genes and Genomes (KEGG) Pathway Enrichment Analyses of the DEGs

GO analysis of 2143 (Control vs. NPs-TiO_2_), 1493 (Control vs. Ethylene), and 1153 (NPs-TiO_2_ vs. Ethylene) DEGs revealed that the 30 most significantly enriched GO terms were in three categories: biological process, cell component, and molecular function. DEGs in the Control and the two treatments encoded enzymes with high catalytic activity, and they were mainly enriched in ‘chloroplast thylakoid membrane’, ‘thylakoid and chloroplast’, and ‘chloroplast’ (Figure 4A,B). DEGs in the NPs-TiO_2_ vs. Ethylene comparison group were mainly enriched in GO terms in the biological process and molecular function categories; these DEGs were also enriched in several GO terms in the cellular component category. Within the molecular function category, the most highly enriched terms were ‘UDP-glucosyltransferase activity’ and ‘enoyl-[acyl-carrier-protein] reductase (NADH) activity’ (Figure 4C).

Next, KEGG pathway enrichment analysis was performed on selected DEGs. DEGs in the Control vs. NPs-TiO_2_ and Control vs. Ethylene comparison groups were significantly enriched in the ‘photosynthesis’ and ‘flavonoid biosynthesis’ pathways (Figure 4D–F). We performed real-time quantitative polymerase chain reaction (qRT-PCR) and transcription fragment per kilobase/million fragment profiling to validate the expression patterns of 12 candidate DEGs enriched in the above KEGG signaling pathways (Appendix A). The expression patterns of nine of the 12 DEGs inferred by the qRT-PCR analyses were consistent with those inferred by the transcriptome analysis.

### 2.5. Differential Expression of Genes and Enzyme Activity Associated with Anthocyanin Biosynthesis in Exocarp

Figure 5 shows the interaction expression of the major genes involved in anthocyanin metabolism in apple peel under different treatments, based on their corresponding positions in the apple anthocyanin biosynthesis pathway for 30 DEGs. The levels of gene expression in fruit treated with both treatments were significantly up-regulated compared to control.

To verify the effects of the NPs-TiO_2_ and ethylene treatments on the expression levels of genes involved in anthocyanin production in apple during light exposure, we examined the expression of eight genes that showed significant changes in expression over the treatment period (Figure 6). The relative expression levels of seven structural genes involved in the anthocyanin synthesis pathway, *MdPAL*, *MdCHS*, *MdCHI, MdF3H*, *MdDFR*, *MdUFGT*, *MdLDOX*, and the key transcription factor *MdMYB1*, are shown in Figure 6A–H. The expression of *MdPAL*, *MdF3H*, *MdLDOX*, and *MdMYB1* first increased and then decreased in the NPs-TiO_2_ and ethylene treatments. The expression of *MdCHS*, *MdCHI, MdDFR*, and *MdUFGT* continuously increased in the NPs-TiO_2_ and ethylene treatment groups.

To clarify the effects of the NPs-TiO_2_ and ethylene treatments on the activities of enzymes involved in anthocyanin synthesis, the activities of three enzymes involved in anthocyanin synthesis were measured. As shown in Figure 6I, significant differences (*p* < 0.05) in phenylalanine ammonia-lyase (PAL) activity were observed in peels treated with NPs-TiO_2_ and ethylene in the early stage of the treatment period (0–24 h). The activity of PAL in the NPs-TiO_2_ treatment increased rapidly at 2 d, and PAL activity was 2.16 times higher at 2 d than at 24 h. The PAL activity was also significantly higher (*p* < 0.05) in the NPs-TiO_2_ treatment than in the ethylene treatment (1.25 times) and control (1.43 times). Changes in the activity of dihydroflavonol 4-reductase (DFR) in NPs-TiO_2_ and ethylene treatments first increased and then decreased, and the activity of DFR was significantly (*p* < 0.05) higher in the two treatments than in the control at 12 h during the light exposure (Figure 6J). The activity of UDP-glucose: flavonoid-3-*O*-glucosyltransferase (UFGT) was higher in the ethylene treatment than in the NPs-TiO_2_ treatment and the control at 24 h; UFGT activity peaked at day 6 and decreased at day 8 (Figure 6K).

### 2.6. Differential Expression of Genes Related to Ethylene and UV Light Signals Pathway

Both ethylene and UV light can promote apple pigment accumulation. We also analyzed the expression of genes involved in the ethylene pathway during light exposure (Figure 7A–D). The results showed that the expression of *MdACS1*, *MdACO1*, *MdERF3* and *MdEIL1* were markedly up-regulated in the ethylene-treated fruit compared to the other two groups. Meanwhile, there were no significant (*p* ≥ 0.05) changes in the expression patterns of ethylene biosynthesis and signaling related genes under NPs-TiO_2_ treatment. We further analyzed the expression levels of genes involved in UV-B signaling (Figure 7E–G). The expression levels of *MdCOP1*, *MdHY5*, and *MdUVR8* in the NPs-TiO_2_ treatment were highest in the early and middle stages of the treatment period (24 h–4 days), and were markedly higher (*p* < 0.05) than those in the other two treatments.

### 2.7. Changes in ROS Levels and Membrane Lipid Peroxidation during Light Exposure

To clarify the effects of the NPs-TiO_2_ and ethylene treatments on oxidative damage to the cell membrane and the degree of membrane lipid peroxidation in the fruit epidermis during light exposure, we measured the cell membrane permeability (Figure 8A) and malondialdehyde (MDA) content (Figure 8B) of the exocarp. At 12 h, the cell membrane permeability of the exocarp treated with the two treatments increased notably (*p* < 0.05), and the cell membrane permeability was 3.07% higher in the ethylene treatment than in the NPs-TiO_2_ treatment. Treatment with NPs-TiO_2_ resulted in a rapid increase in the cell membrane permeability of the peel on day 4, and the cell membrane permeability of the exocarp was significantly higher (*p* < 0.05) in the NPs-TiO_2_ treatment than in the ethylene treatment and control (Figure 8A). The content of MDA increased during the light exposure, and the increase was most pronounced in the ethylene treatment, followed by the NPs-TiO_2_ treatment and the control (Figure 8B); at the end of the treatment period, the content of MDA was markedly higher (*p* < 0.05) in both treatments than in the control.

The total antioxidant capacity of the exocarp tissue during light exposure was determined via a 2,2-diphenyl-1-picryl-hydrazyl-hydrate (DPPH) assay. The content of DPPH in the peel decreased slightly at 12 h and then increased over the treatment period (Figure 8C). The relative content of DPPH was 2.17 times higher in the NPs-TiO_2_ treatment on the eighth day than at the start of the treatment period; the content of DPPH was significantly higher (*p* < 0.05) in the NPs-TiO_2_ treatment than in the ethylene treatment (1.87 times) and control (1.30 times). Changes in the O_2_^−^ production rate and the H_2_O_2_ content were similar over the treatment period (Figure 8D,E). At 24 h, the H_2_O_2_ content was significantly higher in the NPs-TiO_2_ treatment (*p* < 0.05) than in the ethylene treatment and control, and the H_2_O_2_ content increased continuously after 24 h.

The activity of lipoxygenase (LOX) increased in all fruit after 24 h of treatment; LOX activity was significantly higher (*p* < 0.05) in the ethylene treatment than in the NPs-TiO_2_ treatment and control, and LOX activity peaked in the ethylene treatment on the sixth day at 26.04 ± 3.42 U g^−1^. The activity of LOX on the eighth day was 1.27 and 1.95 times (*p* < 0.05) higher in the NPs-TiO_2_ treatment than in the ethylene treatment and control, respectively (Figure 8F). Changes in the activity of catalase (CAT) and superoxide dismutase (SOD) in fruit in the NPs-TiO_2_ treatment were similar over the treatment period, and CAT and SOD activity was higher in the NPs-TiO_2_ treatment than in the ethylene treatment and the control (Figure 8G,H). In the NPs-TiO_2_ treatment group, the activity of SOD peaked (59.80 ± 9.97 U g^−1^) on the fourth day, and the activity of CAT peaked (1603.91 ± 272.66 U g^−1^) on the sixth day.

## 3. Discussion

The appearance quality of fruit has a major effect on its economic value, and dysplasia and instability of the color of fruit during cultivation and production have negative economic effects. NPs-TiO_2_ are semiconductor materials with a band gap of approximately 3.2 eV, corresponding to a wavelength of approximately 390 nm, and the crystal surface of the anatase phase is the most photocatalytically active [33]. Therefore, UV absorption by the apple peel is enhanced after application with NPs-TiO_2_ nano-colloids resulting in higher UV effective absorption. This consequently leads to the accumulation of more pigments. In this study, we found that the effect of the NPs-TiO_2_ treatment on fruit quality was weaker than that of the ethylene treatment. NPs-TiO_2_ treatment had less impact on fruit quality during the accelerated pigmentation process as reflected by the insignificant (*p* ≥ 0.05) reduction in fruit firmness, and the insignificant increase in TSS and the solid–acid ratio. In addition, the ethylene release rate and fruit respiration rate also indicated that the effects of NPs-TiO_2_ treatment alleviated the post-ripening process during the post-harvest treatment period and was weaker compared with the effects of the ethylene spray treatment. These results indicate that NPs-TiO_2_ promoted peel color and had a weaker effect on fruit quality than the ethylene treatment. Light plays an important role in fruit development. Light can affect physiological changes in fruits, especially the synthesis of phenolic substances in the later ripening stage [34]. Primary phenolics in apples are secondary metabolites with antioxidant properties, and the total phenol content and total flavonoid content in some apple varieties are significantly positively correlated with levels of antioxidant activity [35]. Anthocyanins and proanthocyanidins are produced through the flavonoid pathway and are important secondary metabolites that control apple appearance quality, and flavonoids are components of the phenylpropane system [36,37]. Interactions between many important enzymes in the flavonoid pathway are assumed to be the cause of anthocyanin accumulation [38]. In our study, changes in the content of anthocyanins and flavonoids of fruit were more gradual in the NPs-TiO_2_ treatment than in the ethylene treatment, and decreases in the content of proanthocyanidins in the ethylene and NPs-TiO_2_ treatments were similar during the two treatments. This suggests that the activity of the enzymes and transcription factors involved in the flavonoid synthesis pathway were higher under ethylene treatment than in the NPs-TiO_2_ treatment.

Light induces anthocyanin biosynthesis by promoting the expression of key enzyme genes involved in the anthocyanin biosynthesis pathway in apples, and the induction of UV light is extremely important [4,7,8,39]. In this study, KEGG analysis revealed that the DEGs in the two treatments were primarily enriched in ‘Flavonoid synthesis’ pathways, which also corresponded to changes in the content of anthocyanins and flavonoids over the treatment period. This is consistent with the results of previous studies of apple fruits that were treated with UV light, which promoted the accumulation of anthocyanins in the flavonoid metabolic pathway [40,41].

Anthocyanins are derived from the phenylpropanol biosynthesis pathway, and several enzymes are involved in anthocyanin biosynthesis; the functions of anthocyanins have been elucidated [42]. These structural genes are regulated by several transcription factors at the transcriptional and translational levels under different environmental conditions [4]. The proteins encoded by MYB family genes are components of the MYB-bHLH-WD40/WDR protein complex, and MdMYB1 plays an essential role in this process [43,44,45]. We examined seven structural genes involved in anthocyanin production and, with the exception of *MdPAL*, the expression of these genes was higher in the ethylene treatment than in the NP-TiO_2_ treatment. In addition, the transcript levels of the MYB, bHLH, and WD40 group genes were also typically quite different under the two treatments compared to the control. The activity of the three anthocyanin-related enzymes PAL, DFR, and UFGT was determined, and changes in their activity were consistent during the treatment period. Ethylene biosynthesis begins with the synthesis of ACC by 1-aminocyclopropane-1-carboxylate synthase (ACS) from S-adenosyl methionine, which is then oxidized to ethylene by ACC oxidase (ACO). The *ACS* and *ACO* genes have been identified as important factors in the ethylene production in fruit and in the ripening of fruit [46,47]. Ethylene is an important plant hormone, and numerous studies have shown that it can promote anthocyanin accumulation in apple [44,48,49,50]. In apples, *MdEIL1* promotes anthocyanin biosynthesis by directly binding to the *MdMYB1* promoter, and MdERF3 interacts with MdMYB1 to positively regulate ethylene production in the fruits [44]. The expression of genes related to ethylene biosynthesis was remarkably higher (*p* < 0.05) in the ethylene treatment (0.1% ethephon, *w*/*v*) than in the NPs-TiO_2_ treatment and the control. Correspondingly, the ethylene productions were higher in the ethylene treatment, and RNA-Seq data also indicated that the expression levels of structural genes during ethylene biosynthesis were more active in ethylene-treated fruit (Appendix A). All the abovementioned findings showed that ethylene treatment promoted anthocyanin accumulation mainly dependent on the ethylene pathway. In addition, we found that there were no significant changes in the expression patterns of ethylene production and signaling-related genes under NPs-TiO_2_ treatment. UV light can promote apple anthocyanin accumulation [7,8]. Meanwhile, photocatalytic nanomaterials promote fruit coloration by improving the absorption of UV light [30]. Therefore, we further analyzed the expression levels of relevant gene expression in the light signaling pathway. In apple, MdCOP1 and MdHY5 encode key transcription factors involved in anthocyanin biosynthesis and plant photosynthesis [51,52]. In addition, MdUVR8 is a UV-B photoreceptor that can interact with MdCOP1 and plays an essential role in photomorphogenesis [53]. Interestingly, we found the expression levels of these three genes were higher in fruit treated with NPs-TiO_2_ than in fruit treated with ethylene. These findings further indicate that anthocyanin biosynthesis in fruit is induced by UV light signals under NPs-TiO_2_ treatment, which differ from those induced by ethylene.

ROS are signaling molecules that are crucial for controlling plant development, senescence, and growth. Recent investigations have shown that ROS may play a role in controlling the production of anthocyanins in apples [54,55]. According to previous studies, UV radiation stimulates the production of ROS by regulating the activity of NADPH oxidase in the cytoplasmic membrane of the exocarp, thereby enhancing anthocyanin biosynthesis [56,57]. Elevated levels of ROS can induce lipid peroxidation of the cell membrane, which is mainly catalyzed by LOX, and MDA is one of the end products of lipid peroxidation of the cell membrane. Intracellular ROS levels and cell membrane permeability were higher in the NPs-TiO_2_ treatment than in the ethylene treatment and the control during the treatment period. However, the MDA content was slightly higher in the ethylene treatment than in the NPs-TiO_2_ treatment during the treatment period; a sharp increase in LOX enzyme activity was observed at 4 days and 6 days. We evaluated levels of antioxidant activity in fruit during the treatment period and found that the levels of antioxidant enzymes and the DPPH content were higher in the NPs-TiO_2_ treatment than in the ethylene treatment and the control during the treatment period. This also explains why the MDA content was lower in the NPs-TiO_2_ treatment group than in the ethylene treatment group. In sum, treatment with NPs-TiO_2_ and ethylene induced the accumulation of anthocyanins in fruit during light exposure. The antioxidant capacity and ROS levels of fruits were higher and levels of membrane lipid peroxidation were lower in the NPs-TiO_2_ treatment than in the ethylene treatment. Additional research is needed to determine whether the ROS generated via the photocatalytic activity of NPs-TiO_2_ or the absorption and use of UV light by NPs-TiO_2_ are the principal drivers of anthocyanin biosynthesis in fruit.

## 4. Materials and Methods

### 4.1. Preparation and Characterization of NPs-TiO_2_

NPs-TiO_2_ were thermally synthesized following the methods described in previous studies [30,58]. The prepared NPs-TiO_2_ were characterized by scanning electron microscopy (SEM), transmission electron microscopy (TEM), dynamic light scattering (DLS) and X-ray crystal diffraction (XRD). The obtained NPs-TiO_2_ had an irregular spherical shape and a crystal conformation with a particle size of 10.34 ± 3.34 nm (Appendix A). After obtaining dry NPs-TiO_2_ powder, NPs-TiO_2_ colloidal solution was prepared via an ultrasonic process [30].

### 4.2. Fruit Material and Experimental Treatments

Apple fruit (*Malus domestica* Borkh. ‘Red Delicious’) was collected from an orchard in Tai‘an, Shandong Province, China. The fruits were bagged at the young fruit stage (40 days after the flowering), and the orchard was managed using standard practices during growth and development. The fruits used in this study were harvested at 150 days after flowering (DAF). Apples of the same size and with no obvious signs of physiological disease were collected and stored in soft-bottom foam boxes. They were then transported to the laboratory of Shandong Agricultural University the same day that they were collected for subsequent experimental treatments. The apples were divided into three groups, with 30 apples in each group, and transferred into bags in low-light environments. Following previous studies [44,59], 0.1% (*w*/*v*) ethephon solution and 0.1% (*w*/*v*) NPs-TiO_2_ colloids were sprayed on apples; in the control treatment, the same amount of deionized water was applied. After the surface water dried naturally, the cells were placed into three independent light incubators (25 °C, 70 μmol m^−2^ s^−1^ intensity of illumination. and 330 ± 20 nm wavelength) for light cultivation for 8 days. The peel tissue was sampled at 0 h, 12 h, 24 h, 2 days, 4 days, 6 days and 8 days. Three fruit skin samples were taken from each treatment group at each time period and discarded immediately after the skin samples were taken. The samples were rapidly frozen with liquid nitrogen and stored at −80 °C in a refrigerator for subsequent experiments.

### 4.3. Determination of Fruit Physiological Quality

The exocarp color of the apple was determined using a chroma meter at the indicated times (0 h,12 h, 24 h, 2 days, 4 days, 6 days and 8 days). Fruit treated with light for 8 days were measured for firmness, total soluble solids (TSS), total titratable acid (TA) and solid–acids ratio physiological qualities as described in the previous study [59]. The respiration rate and ethylene production of the experiment fruit at the indicated times (0 h, 4 days, 8 days) were measured using a CO_2_ detector (Testo535-CO_2_, Testo Inc., Shanghai, China) and a gas chromatograph (GC-2014, Shimadzu, Toyko, Japan), respectively, as described in previous studies [60]. A total of six biological replicates were performed for each treatment in the above experiments.

### 4.4. Determination of Secondary Metabolites

The anthocyanin content in exocarp was determined using the hydrochloric acid–methanol method with reference to the previous study [61]. The supernatant’s absorbance was calculated at 530, 620, and 650 nm. Anthocyanin content was expressed as 0.1 change in optical density (μg g^−1^ FW).

The content of proanthocyanidins was determined according to the method of previous research with slight modification [62]. At a wavelength of 640 nm, the absorbance readings of reaction system were recorded. The standard curve was made of catechin (μg g^−1^ FW).

The flavonoids and the total phenolics content were determined with reference to previous studies [63]. For the flavonoids content, the absorbance value of reaction solution at 430 nm wavelength was recorded. The rutin-based standard curve was used (μg g^−1^ FW). For the total phenolics content, the absorbance value of reaction solution at 725 nm wavelength was recorded. The gallic-acid-based standard curve was used (μg g^−1^ FW).

### 4.5. RNA-Seq Analysis

RNA was extracted from exocarp tissue exposed to 8 days of light, and purified RNA was collected in triplicate for each treatment group. OE Biotech Co., Ltd. carried out the sequencing and analysis of the transcriptome (Shanghai, China). Trimmomatic was used to process the raw data [64]. To achieve the clean reads, the low-quality reads and reads containing ploy-N were eliminated. A total of 60.47 G of clean data was obtained. Effective data for each sample ranged from 6.57G to 6.96 G, with Q30 bases ranging from 91.2% to 91.98%, and GC fractions ranging from 44.45% to 47.62% (Appendix A). Following that, hisat2 was used to map the clean reads to the reference genome [65]. By using the approaches from previous research [66,67,68,69,70], quantitative analysis of gene expression, differentially expressed genes analyses (DEGs), Gene Ontology analyses (GO) and Kyoto Encyclopedia of Genes and Genomes (KEGG) enrichment analyses were carried out. The threshold for significantly altered expression was chosen at *p* < 0.05 and foldchange > 2 or foldchange < 0.5. The RNA-seq raw data were submitted to NCBI with the following ID number: PRJNA947803.

### 4.6. Real-Time qPCR Expression Analysis

RNA plant plus Reagent kit (Tiangen, Beijing, China) was used to extract RNA from apple exocarp tissue according to the corresponding experimental methods. After obtaining the RNA of the pure sample, the PrimeScript cDNA synthesis kit (Tiangen, Beijing, China) was used for subsequent reverse transcription experiments to obtain the cDNA template of the sample. The cDNA template was subsequently diluted to a concentration of approximately 2 ng μL^−1^. The qRT-PCR was performed using the 2^−ΔΔCt^ calculation method, and the results were normalized using the *MdActin* gene of apple as the reference gene. The qRT-PCR procedures were set according to the experimental methods of previous studies [52]. The primer sequences for all genes are provided in Appendix A.

### 4.7. Determination of Malonaldehyde (MDA) Content, Cell Membrane Permeability and Reactive Oxygen Species (ROS) Levels

The relative cell membrane permeability of the exocarp tissue was detected using an electrical conductivity meter (DDS-307A, INESA Scientific Inc., Shanghai, China) at the indicated times (0 h, 12 h, 24 h, 2 days, 4 days, 6 days and 8 days). The relative cell membrane permeability is expressed as % [60].

The content of MDA in exocarp was determined by trichloroacetic acid (TBA) method with slight modification according to the previous study [71]. The sample’s absorbance value at wavelengths of 600 nm, 532 nm, and 450 nm was calculated (μmol g^−1^ FW).

The production rate of superoxide anion and the content of hydrogen peroxide (H_2_O_2_) in exocarp were determined by Superoxide Anion kit and Hydrogen Peroxide detection kit (Suzhou Keming Biotechnology Co., Ltd., Suzhou, China), respectively. The absorption value of the reaction system at a wavelength of 530 nm was recorded by a microplate reader to determine the superoxide anion production rate. One hour passed without any static reaction occurring in the centrifuge tubes of the control group. The result is denoted as nmol g^−1^ min^−1^ FW. For H_2_O_2_ content, the absorbance value of reaction solution at 415 nm wavelength was recorded (μmol g^−1^ FW).

### 4.8. Determination of Total Antioxidant Capacity

The scavenging activity of 2,2-diphenyl-1-picryl hydrazide (DPPH) free radical in exocarp tissues was measured by using the previous method [72]. The same mass fraction of ethanol solution was used to generate the standard solution curve of Trolox. At a wavelength of 515 nm, the reaction solution’s absorbance was recorded. The results of DPPH relative content are expressed as μmol Trolox g^−1^ FW.

### 4.9. Determination of Enzyme Activity

For LOX, one gram of frozen materials was crushed in 10 mL of 0.05 mM of phosphatic buffer solution (PBS, PH 7.8, containing 50 mM β-mercaptoethanol), and the homogenate was centrifuged for 10 min (4 °C, 12,000 rpm). The substrate for the enzymatic reaction was prepared by adding 0.5 mL linoleic acid, 0.25 mL Tween 20, and 1 mL NaOH (1 M) to 20 mL borate buffer, followed by adjusting the pH to 9.0 and diluting to 500 mL. The reaction system was prepared by mixing 0.2 mL of the supernatants with 0.8 mL of the enzyme substrates [73]. The absorbance value of reaction system at 234 nm wavelength was detected using a microplate reader (U g^−1^ FW)

The extraction and determination methods of superoxide dismutase (SOD) and catalase (CAT) were referred to previous studies [63]. The ability of the samples to inhibit the photo-chemical reduction of Nitro-Blue Tetrazolium (NBT) was analyzed to assay the enzyme activity of SOD. The absorbance value of reaction solution at 560 nm wavelength was detected by microplate reader (U g^−1^ FW). For CAT, the absorbance value of reaction solution at 240 nm wavelength was detected by microplate reader. A unit of enzyme activity was defined as 1 nmol H_2_O_2_ degradation per minute per gram of tissue catalyzed (U g^−1^ FW).

The enzyme activities of l-Phenylalanine Ammonia-Lyase (PAL), Dihydroflavonol 4-Reductase (DFR) and UDP-glycose flavonoid glycosyltransferase (UFGT) enzyme activity were determined by enzyme activity detection kit (Hefei Laier Biotechnology, Inc., China). For PAL, the absorbance of reaction solution at a wavelength of 290 nm was measured using a microplate reader. Each gram of tissue in each milliliter reaction system changes the absorbance value at 290 nm by 0.1 per minute, which is one unit of enzyme activity (U g^−1^ FW). For DFR, the absorbance value of reaction solution at a wavelength of 500 nm was measured using a microplate reader. A unit of enzyme activity was defined as 1 mmol catechin per gram of tissue per minute of tissue catalyzed (U g^−1^ FW). For UFGT, the absorbance value of reaction solution at 340 nm wavelength was detected by microplate reader (U g^−1^ FW). A unit of enzyme activity was defined as 1 nmol NADH per minute per gram of tissue catalyzed.

### 4.10. Statistical Analysis

The experiment was conducted in a completely random manner. The data were evaluated by One-way analysis of variance (ANOVA) using Excel 2019 and Origin 2021 software. At least three biological replicates were used in each experiment. All data were reported as mean ± standard error (SE). The ANOVA was performed by *p* < 0.05. The difference was significant by least partial square test (LSD).

## 5. Conclusions

We analyzed the effects of treatment with NPs-TiO_2_ and ethylene on the anthocyanin biosynthesis and ROS metabolism in ripe ‘Red Delicious’ apples under light exposure. The application of both treatments significantly promoted the deposition of pigments in the peel during light exposure, and NPs-TiO_2_ had a minor effect on the quality of the fruit during the treatment period. Furthermore, ethylene treatment promoted fruit coloring mainly by up-regulating the expression of ethylene-related genes, whereas NPs-TiO_2_ promoted fruit coloring mainly by up-regulating the expression of UV-radiation-related genes. The ROS levels and antioxidant capacity of the treated fruit were higher in the NPs-TiO_2_ treatment group than in the ethylene treatment group during the treatment period, and the degree of membrane lipid peroxidation was lower in fruit in the NPs-TiO_2_ treatment than in the ethylene treatment. These results imply the potential utility of NPs-TiO_2_ in enhancing apple coloration. Moreover, the transcriptomic data offer novel insights into the molecular mechanisms underlying the promotion of apple coloration via NPs-TiO_2_ treatment.

## Figures and Tables

**Figure 1 foods-12-03137-f001:**
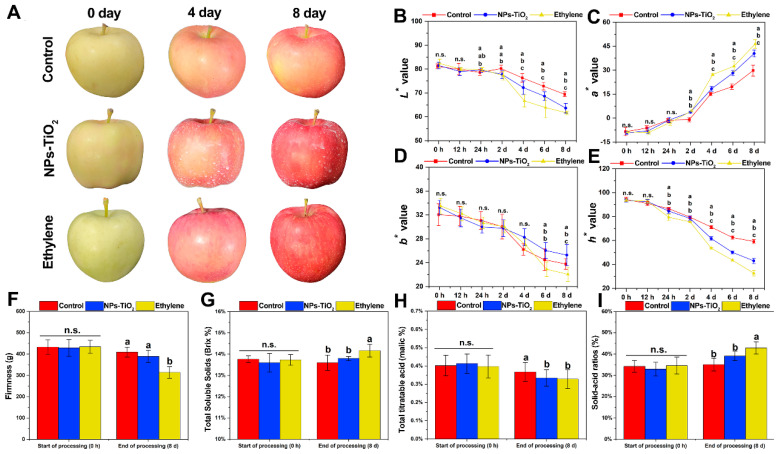
Effects of NPs-TiO_2_ and ethylene treatments on color and fruit physiological quality of “Red Delicious” apple during light exposure. (**A**) Fruit coloring phenotype after treatment, (**B**) *L** value (black to white), (**C**) *a** value (green to red), (**D**) *b** value (yellow to blue), (**E**) *h** value (hue angle), (**F**) Firmness, (**G**) Total soluble solids (TSS), (**H**) Total titratable acid (TA), and (**I**) Solid–acid ratios (TSS/TA ratio). Error bars, mean ± standard error (*n* = 6). Different letters indicate significant differences among treatments in the same period using Duncan’s test (*p* < 0.05), n.s. represents no significant difference (*p* ≥ 0.05).

**Figure 2 foods-12-03137-f002:**
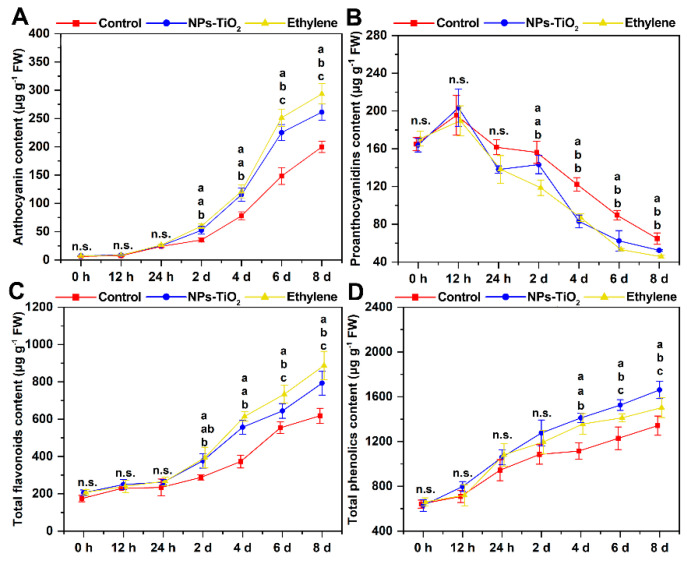
Effects of two treatments on (**A**) anthocyanin content, (**B**) proanthocyanidin content, (**C**) total flavonoids content and (**D**) total phenolics content during light cultivation of ‘Red Delicious’ apples. Error bars, mean ± standard error (*n* = 3). Different letters indicate significant differences among treatments in the same period using Duncan’s test (*p* < 0.05), n.s. represents no significant difference (*p* ≥ 0.05).

**Figure 3 foods-12-03137-f003:**
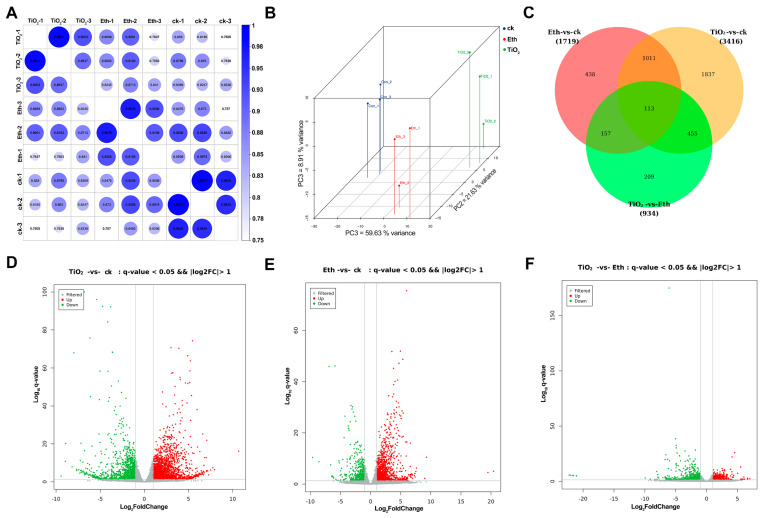
Differentially expressed genes (DEG) in apple after NPs-TiO_2_ and ethylene treatment. (**A**) Correlation heatmap of different treatment groups, (**B**) PCA score diagram of DEGs, (**C**) Venn diagram showing the shared and distinct DEGs among the three compared groups, (**D**–**F**) Volcano plots displaying the up-regulated and down-regulated genes between different treatment groups. ck represents Control, TiO_2_ represents NPs-TiO_2_, and Eth represents Ethylene.

**Figure 4 foods-12-03137-f004:**
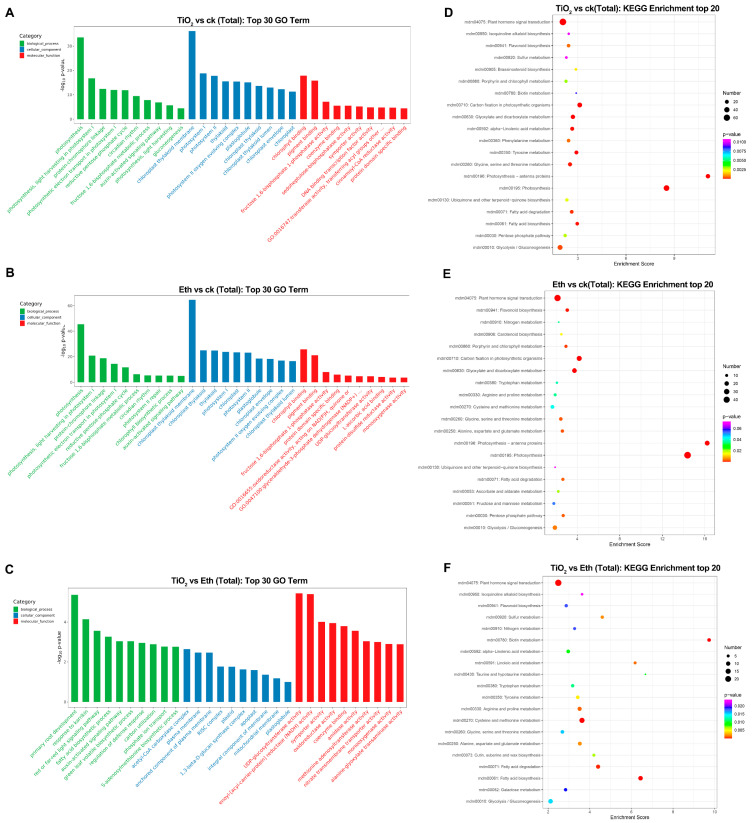
GO classification and KEGG pathway enrichment analyses of differentially expressed genes (DEGs) under the different treatments. Top 30 GO classification analysis of the genes identified under the biological process, molecular function, and cellular component categories between (**A**) NPs-TiO_2_ vs. Control, (**B**) Ethylene vs. Control and (**C**) NPs-TiO_2_ vs. Ethylene. Top 20 KEGG enrichment analysis of the DEGs between (**D**) NPs-TiO_2_ vs. Control, (**E**) Ethylene vs. Control and (**F**) NPs-TiO_2_ vs. Ethylene. Number of DEGs is represented by the size of the circle. ck represents Control, TiO_2_ represents NPs-TiO_2_, and Eth represents Ethylene.

**Figure 5 foods-12-03137-f005:**
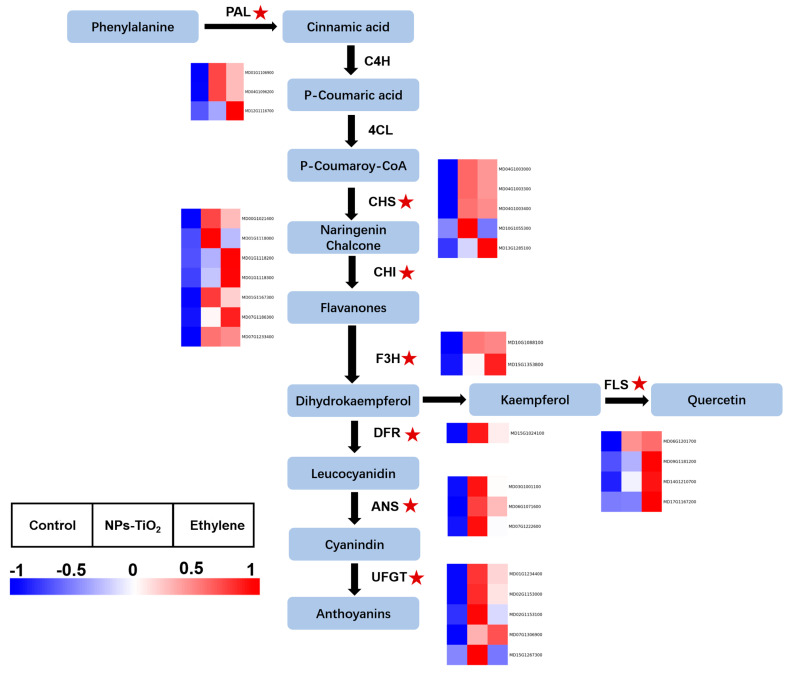
The regulation of flavonoids and anthocyanins biosynthesis pathways in apple under different treatments during light exposure. Heatmaps of DEGs in transcripts from flavonoids and anthocyanins biosynthesis pathway. PAL, phenylalanine ammonia-lyase; CHS, chalcone synthase; CHI, chalcone isomerase; F3H, flavanone-3-*O*-hydroxylase; DFR, dihydroflavonol 4-reductase; ANS, anthocyanidin synthase; UFGT, UDP glucose: flavonoid-3-*O*-glucosyltransferase; FLS, flavonoid 3′-monooxygenase. Red five-pointed stars represent significant DEGs in signaling pathways.

**Figure 6 foods-12-03137-f006:**
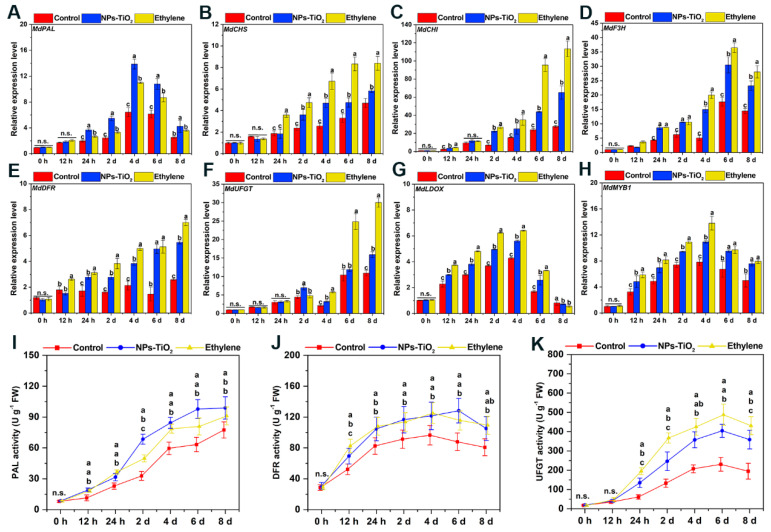
Expression of genes (**A**–**H**) and enzyme activities related to anthocyanin biosynthesis (**I**–**K**) in the ‘Red Delicious’ apple exocarp tissues during light exposure with NPs-TiO_2_ and ethylene treatments. Error bars, mean ± standard error (*n* = 3). Different letters indicate significant differences among treatments in the same period using Duncan’s test (*p* < 0.05), n.s. represents no significant difference (*p* ≥ 0.05). PAL, phenylalanine ammonia-lyase; CHS, chalcone synthase; CHI, chalcone isomerase; F3H, flavanone-3-*O*-hydroxylase; DFR, dihydroflavonol 4-reductase; UFGT, UDP glucose: flavonoid-3-*O*-glucosyltransferase; LDOX, leucoanthocyanidin dioxygenase; MYB, R2R3-MYB transcription factor.

**Figure 7 foods-12-03137-f007:**
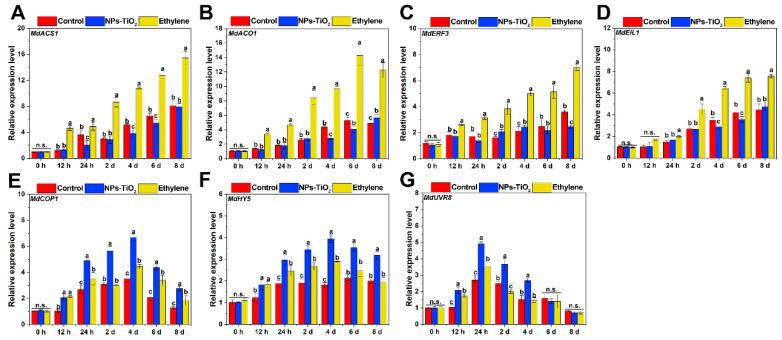
Expression of genes related to ethylene (**A**–**D**) and UV light signals pathway (**E**–**G**) in the ‘Red Delicious’ apple exocarp tissues during light exposure with the NPs-TiO_2_ and ethylene treatments. Error bars, mean ± standard error (*n* = 3). Different letters indicate significant differences among treatments in the same period using Duncan’s test (*p* < 0.05), n.s. represents no significant difference (*p* ≥ 0.05). ACS1, 1-aminocyclopropane-1-carboxylate synthase 1; ACO1: 1-aminocyclopropane-1-carboxylate oxidase 1; EIL1: ethylene insensitive3-like 1; ERF3: ethylene response factors 3; COP1, constitutively photomorphogenic 1; HY5, elongated hypocotyl 5; UVR8, UV resistance locus 8.

**Figure 8 foods-12-03137-f008:**
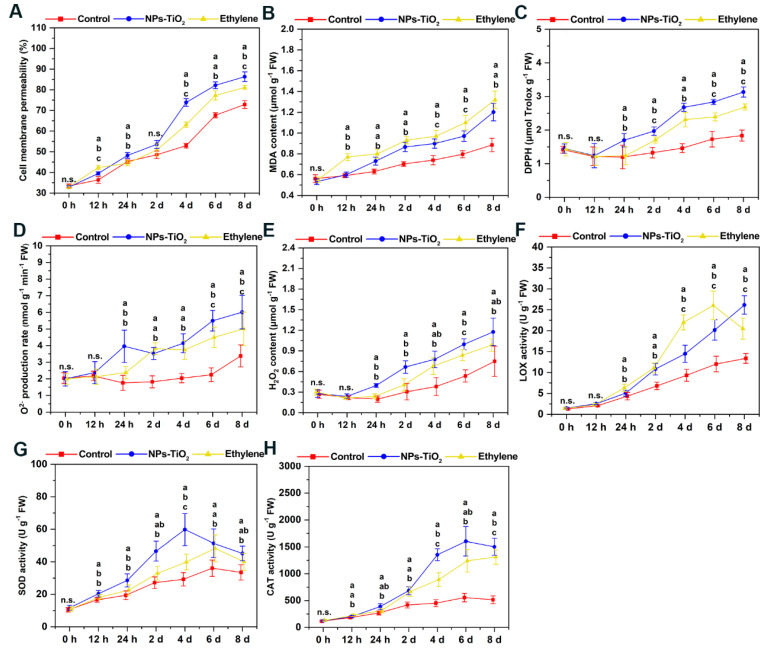
Effects of two treatments on (**A**) Cell membrane permeability, (**B**) MDA content, (**C**) DPPH, (**D**) O_2_^−^ production rate and (**E**) H_2_O_2_ content, (**F**) LOX activity, (**G**) SOD activity and (**H**) CAT activity during light cultivation of the ‘Red Delicious’ apples. Error bars, mean ± standard error (*n* = 3). Different letters indicate significant differences among treatments in the same period using Duncan’s test (*p* < 0.05), n.s. represents no significant difference (*p* ≥ 0.05).

## Data Availability

The data used to support the findings of this study can be made available by the corresponding author upon request.

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
