# Peer review of "Post-Harvest Application of Nanoparticles of Titanium Dioxide (NPs-TiO2) and Ethylene to Improve the Coloration of Detached Apple Fruit"

_foods, 2023, doi:10.3390/foods12163137_

Round 1

Reviewer 1 Report

The manuscript describes thoroughly the effect of two coloring methods of apple fruit with aim of explaining the mechanisms behind.

Specific comments:

When reading ck remains unexplained until the material and method section. Suggest avoid the abbreviation in figures and text. Control is a relatively short word.

L15 less effect on fruit quality should be explained. Do you mean fruit color?

Figure 1. Phenotype? Is that treatment? TSS is a know abbreviation also TA but solid-acid is not ok, could use TSS/TA ratio. Give cultivar namn.

Figure 2. Include cultivar name. Avoid ck as abbreviation. The last part, The LSD test at a P value.. must be unnecessary.

Figure 3. Why preliminary? The legend lack details of what it is analysed.

Figure 4. The legend lack details of what the experiment is, apple etc.

Figure 5 avoid the abbrevations for treatments

Figure 6 include apple fruit, explain abbreviations

Figure 7 include apple fruit, explain abbreviations

Use fruit instead of fruits when explain about more than one apple fruit. E.g. in line 375. The fruit were bagged, not fruits. Fruits should be used when having more than one specie.

L377 harvested at the fruit ripening stage is not a scientific correct term. Be more concrete.

Author Response

The manuscript describes thoroughly the effect of two coloring methods of apple fruit with aim of explaining the mechanisms behind.

Thank you for your careful review of our research and your valuable comments. We attach great importance to the problems you pointed out, and will modify and improve according to your suggestions.

Specific comments:

When reading ck remains unexplained until the material and method section. Suggest avoid the abbreviation in figures and text. Control is a relatively short word.

Response: Thank you for your valuable feedback. On the question of abbreviations, we will follow your advice to ensure that abbreviations are fully explained in the manuscript and to avoid using abbreviations in the Figures and text. In addition, we use the full form of "Control" where appropriate to increase clarity and legibility of the text.

L15 less effect on fruit quality should be explained. Do you mean fruit color?

Response: Thank you for your valuable feedback. ​I apologize for making this part unclear. The effect of NPs-TiO2 on fruit firmness, TSS, TA, and TSS/TA ratio is smaller than that of ethylene treatment. We have corrected line17-19 of the manuscript.

Figure 1. Phenotype? Is that treatment? TSS is a know abbreviation also TA but solid-acid is not ok, could use TSS/TA ratio. Give cultivar name.

Response: Thank you for your valuable feedback. ​Figure 1A shows the fruit phenotypes at 0, 4, and 8 days of light culture after treatment with different colorants. We modify the legends and text about Figure 1 and reinterpret them.

​The descripts of the total soluble solids (TSS), the total titratable acid (TA) and solid-acid ratios (TSS/TA ratio), and the figure legends have been corrected in the manuscript as requested. The apple cultivar name has been added in line 84-88 in the manuscript.

Figure 2. Include cultivar name. Avoid ck as abbreviation. The last part, The LSD test at a P value. must be unnecessary.

Response: Thank you for your valuable feedback. We have revised the legends section of Figure 2 to add the apple variety name and remove redundant statistics notes (The LSD test at a P value of 0.05 was used).

Figure 3. Why preliminary? The legend lack details of what it is analysed.

Response: Thank you for your valuable feedback. ​It refers to the preliminary analysis of raw transcriptome data in order to find differential expressed genes (DEGs) for different treatment groups. To avoid ambiguity we have modified the legend of Figure 3 according to the manuscript (in line 141-146).

Figure 4. The legend lack details of what the experiment is, apple etc.

Response: Thank you for your valuable feedback. ​We have revised and added the corresponding details to the legends of Figure 4 (in line 166-173).

Figure 5 avoid the abbrevations for treatments

Response: Thank you for your valuable feedback. ​We have changed the abbreviation of the treatments in Figure 5 to the full name.

Figure 6 include apple fruit, explain abbreviations

Response: Thank you for your valuable feedback. We have added the full names of the relevant abbreviations in figure 6 legends in line 214-217 of the manuscript.

Figure 7 include apple fruit, explain abbreviations

Response: Thank you for your valuable feedback. We have added the full names of the relevant abbreviations in figure 7 legends in line 249-253 of the manuscript.

Use fruit instead of fruits when explain about more than one apple fruit. E.g. in line 375. The fruit were bagged, not fruits. Fruits should be used when having more than one specie.

Response: Thank you for your valuable feedback. Regarding ‘fruit’ and ‘fruits’, we have revised them all in the manuscript.

L377 harvested at the fruit ripening stage is not a scientific correct term. Be more concrete.

Response: Thank you for your valuable feedback. We have rewritten that about the harvest time (in line 428-429).

We genuinely value your constructive feedback, and we are committed to making the necessary revisions to improve the manuscript significantly. We are confident that the revised version will meet the journal's standards and contribute meaningfully to the field of apple coloration research. Thank you once again for your time and guidance. We look forward to resubmitting the revised manuscript for your consideration.

Reviewer 2 Report

see attachment

see attachment

Author Response

The manuscript by Chinese authors is on apple colouration using a comparison between nanoparticle tioxide and an ethylene releasing compound. The topic fits the scope of the journal foods. The manuscript is confusing, because the reader thinks the apples are still on the tree ("attached”) and some statements in the text are incorrectly referenced- the references do not match the statements and there are few formal issues with line and page breaks. The manuscript also lacks an explanation how the TiOx is washed off the fruit. Overall, the manuscript can be rejected, because it would require major revision of the following many issues.

Response: Thank you for taking time out of your busy schedule to review this paper on apple coloring. We acknowledge the issues raised in your review, and we are committed to addressing them thoroughly. Specifically, we will work on the following points:

  • Clarifying the Experimental Setting: We understand that the manuscript may give the impression that the apples are still "attached" to the tree during the experiments.
  • Correcting Referencing Errors: We apologize for any inaccuracies in referencing within the text. We will carefully cross-check the references and ensure that they properly correspond to the statements they support.
  • Addressing Formal Issues: We will pay close attention to line and page breaks to ensure the manuscript's readability and formatting meet the journal's requirements.
  • Explaining NPs-TiO2 Washing Procedure: We understand the importance of providing a thorough explanation of how the NPs-TiO2 is washed off the fruit. We will include a detailed procedure to address this gap in the manuscript.

Title

1) Title is a) wrong, b) too long and c) too clumpsy- try e.g. " Post-harvest application of nanoparticle tioxide (NP-Ti-ox) and ethylene can improve colouration of detached apple fruit'- Ethephon is the name of a commercial product- trade name- so use the scientific name ethylene. The old title is very confusing, as many researchers try to spray Tioxide on the ground to improve light reflection and red skin in apple in the orchard.

Response: Thank you for your valuable feedback on the title of our manuscript. We truly appreciate your insights, and we have carefully considered your suggestions.

Revised Title:

“Post-harvest application of nanoparticles titanium dioxide (NPs-TiO2) and ethylene on improving colouration of detached apple fruit”

2) Keywords - add nanoparticle(s)

Response: Thank you for your valuable feedback. This keyword has been added to the revised manuscript.

3) Abstract- line 14 insert “detached mature” before apple and an explanation that the fruit had been bagged before treatment

Response: Thank you for your valuable feedback. The abstract of the article has been revised as suggested in line 18 of the manuscript. The bagging of fruit is described in the Materials and methods section (in line 426-427).

4) The headings of Abstract and Keywords are too large.

Response: Thank you for your valuable feedback. We have removed the "fruit quality" in the keyword and modified the abstract (in line 29-31).

Introduction

5) Line 30, reference (1] on the apple genome is wrong- you need FAO and Chinese statistics and

you need to say that apple production worldwide is ca 82 mil tons with China exceeding 42 mil tons or such like plus reference for 2022.

Response: Thank you for your valuable feedback. We apologize for the incorrect reference and the missing information in line 30 regarding apple production worldwide. We have rewritten this part and removed this reference, with the revised section at line 35-37 of the manuscript.

6) line 31, add the consumer preference, which is missing here and important- the consumer associates red (the red colour of tomato or apple fruit) with ripeness and sweet taste.

Response: Thank you for your valuable feedback. We are rewriting this part as suggested and cited with relevant reference. The revision is in line 37-39 of the manuscript.

7) Lines 33-34, close up gap /line break

Response: Thank you for your valuable feedback. ​We apologize for the line break in the original manuscript, which created a gap in the text. It has been revised in the newly revised manuscript.

8) Line 35, add "in combination with low temperature" (ref- e.g. Funke-https://doi.org/10.3390/horticulturae7010002)- you need PAR plus UV plus low temperature for apple fruit colouration.

Response: Thank you for your valuable feedback. We have studied and cited the reference you provided, and the corresponding part of the manuscript has been revised (line 42-43).

9) Line 40-41, substitute the term “drugs” by “biostimulants or such like”

Response: Thank you for your valuable feedback. We have revised this word in the manuscript as suggested, in line 49 of the manuscript.

10) Line 36- you need to say/add that it is the combination of PAR and UV and cite e.g. Weber et al 2019 (DOI:10.1016/j.jplph.2018.12.008)

Response: Thank you for your valuable feedback. We have studied and cited the reference you provided, and the corresponding part of the manuscript has been revised (line 42-43).

11) Line 41, again references 6 and 7 refer to apple fruit ripening, NOT to the range of means to improve colour- refer to papers of Weber (2019) or Funke (2021) or other authors;

Response: Thank you for your valuable feedback. We have studied and cited the reference you provided, and the corresponding part of the manuscript has been revised (line 50-52).

12) Lines 45-46, substitute “chemical colorants” by “ethylene releasing compounds”

Response: Thank you for your valuable feedback. We have revised this word in the manuscript as suggested, in line 57-58 of the manuscript.

13) Line 61, please double-check whether this should be “exocarp”, not pericarp- Pericarp is usually the “flesh” of the apple fruit, exocarp the “skin”- you use epidermal correctly in line 77

Response: Thank you for your valuable feedback. We have substituted “pericarp” by “exocarp” in the revised manuscript.

14) Line 62, You need to mention that kaolin products like SurroundTM:' are used for the opposite, prevent light penetration into the skin and combat sunburn ref) like e.g. Grange 2004 -https://doi.org/10.17660/ActaHortic.2004.636.69.

Response: Thank you for your valuable feedback. We have studied and cited the reference you provided, and the corresponding part of the manuscript has been revised (in line 73-74).

Results and Discussion

15) Figure 1 B, I think “brightness” is the better word than “lightness” (which could also be interpreted as less weight, and in the text lines 77-80

Response: Thank you for your valuable feedback. We have modified the legend and pictures in Figure 1 to avoid ambiguity. We have changed “lightness” to “brightness” in the manuscript (in line 95).

16) Figure 1 C- change to:” a value (green to red), D b value (yellow to blue, both in the CIE colour

scheme)” and in the text 77-80, where you say yellowness, but blueness in figure 1

Response: Thank you for your valuable feedback. We have revised the titles of vertical axis in Figure C, D as suggested, and the legend have also been modified accordingly in line 84-88 of the manuscript.

17) Figure 1- write out TSS and TA in full – like in the text

Response: Thank you for your valuable feedback. We have written out TSS and TA in full in the legends of Figure1 and text as you suggested.

18) The manuscript lacks an explanation how the TiOx is washed off the fruit.

Response: Thank you for your valuable feedback.

We apologize for the confusion caused by this issue, in our previous study, it was found that NPs-TiO2 attached to the surface of the peel could be washed off by simple cleaning. NPs are insoluble in water, and because they are chemically stable, they do not form stable chemical bonds (covalent bonds) when they come into contact with the fruit epidermis.

Although NPs-TiO2 nanoparticles are chemically stable and non-toxic, they are not absorbed and utilized by the human body. Therefore, we designed a wash test for apples treated with NPs-TiO2 nanoparticles at that time. Scanning electron microscopy and EDS elemental analysis showed that no NPs-TiO2 remained on the washed fruit surface. ​Furthermore, we observed that apple peel with dense cuticle effectively prevents NPs-TiO2 particles from entering the pulp. To avoid confusion here, we have made changes in the manuscript (in line 70-73).

Reference: Wang C, Wang Y, Sun Q, et al. Nanoscale UV absorber boosting coloration of apple fruit skin. ACS Sustainable Chemistry & Engineering, 2019, 7(19). DOI:10.1021/acssuschemeng.9b03316.

References

19) references 7, 11 and others: Remove given (Christian) names

Response: Thank you for your valuable feedback. We have removed given names in references 13, 29, 36, 37 in the revised manuscript.

20) Saure needs his given name “Max” => to read Saure, M., xxxxxx

Response: Thank you for your valuable feedback. We have corrected the information in this reference.

Format

21) Close up bottom of page 8 and delete page 18 – which are empty

Response: Thank you for your valuable feedback. We have closed up bottom of page 8 and delete page 18.

We genuinely value your constructive feedback, and we are committed to making the necessary revisions to improve the manuscript significantly. We are confident that the revised version will meet the journal's standards and contribute meaningfully to the field of apple coloration research. Thank you once again for your time and guidance. We look forward to resubmitting the revised manuscript for your consideration.

Reviewer 3 Report

Review of the article «Insights into the differences in apple coloring and ROS metabolism regulated by NPs-TiO2 and ethephon colorants»

 The article is interesting and deals with a subject that may be of interest to fruit growing. However, there are some shortcomings in the article.

Title: The word "colorant" should be deleted, because ethephon is not a colorant

Line 30: The cited article ([1]) is not an article about apple production and consumption. In this work it is written that “The domesticated apple (Malus × domestica Borkh., family Rosaceae, tribe Pyreae) is the main fruit crop of temperate regions of the world.”, but the article does not cite a source, nor is it the result of a work on apple production and consumption. Information on the importance of crops can be taken from the FAO database or from articles on the production and market of each crop. In this case, authors should choose to cite FAO data or an article on apple production and market, like this one, for example: doi: 10.17660/eJHS.2022/059. This article also talks about the increased importance of red skin cultivars.

Line 33: There is an accidental line change that needs to be eliminated.

Line 34: References 2 and 3 do not seem very appropriate to me. These articles talk about trends in food, but do not have the nutritional values of fruits. They don't even talk about apples.

Figure 1: I assume "ck" is the control, but it should be spelled out in both the graphic and the text. In this case, the abbreviation saves almost no space and makes it difficult to read, even if the meaning of the abbreviation is explained the first time it is used.

Figure 1D; Line 74: “blueness” should be replaced by “Yellowness”, like in the graph.

Lines 83-84: The hue angle does not indicate the amount of accumulated pigments, but the type of pigments. This parameter is a hue (color) indicator. The color intensity (chroma) could be considered an indicator of the amount of pigments. Unfortunately, this parameter (chroma) is not presented.

Lines 85: I assume "8 d" means "8 days". To save 3 characters, it made me spend several seconds looking for an 8d figure, until I understood the meaning of the abbreviation. If authors want readers to read their article easily and with pleasure they should avoid excessive use of abbreviations.

4. Materials and Methods: The authors do not adequately describe the process the fruits went through, nor the exact way in which cell samples were taken. Cells were taken from how many fruits? Were the fruits from which the cells were removed discarded, or did they follow the procedures after being “mutilated”?

Lines 508-509: I think the conclusion is too affirmative, even with regard to the apple tree. It is still much more exaggerated when trying to conclude that the treatment can be applied to all horticultural crops.

I attach a file with some comments and marked text

English is not my mother tongue. From what I can judge, the authors should make some improvements to the English of the article, but I don't think it's too bad.

Author Response

Review of the article «Insights into the differences in apple coloring and ROS metabolism regulated by NPs-TiO2 and ethephon colorants»

The article is interesting and deals with a subject that may be of interest to fruit growing. However, there are some shortcomings in the article.

Response: Thank you for taking time out of your busy schedule to review this paper on apple coloring. We acknowledge the issues raised in your review, and we are committed to addressing them thoroughly.

Title: The word "colorant" should be deleted, because ethephon is not a colorant

Response: Thank you for your valuable feedback on the title of our manuscript. We truly appreciate your insights, and we have carefully considered your suggestions.

Revised Title:

“Post-harvest application of nanoparticles titanium dioxide (NPs-TiO2) and ethylene on improving colouration of detached apple fruit”

Line 30: The cited article ([1]) is not an article about apple production and consumption. In this work it is written that “The domesticated apple (Malus × domestica Borkh., family Rosaceae, tribe Pyreae) is the main fruit crop of temperate regions of the world.”, but the article does not cite a source, nor is it the result of a work on apple production and consumption. Information on the importance of crops can be taken from the FAO database or from articles on the production and market of each crop. In this case, authors should choose to cite FAO data or an article on apple production and market, like this one, for example: doi: 10.17660/eJHS.2022/059. This article also talks about the increased importance of red skin cultivars.

Response: Thank you for your valuable feedback. We apologize for the incorrect reference and the missing information regarding apple production worldwide. ​We have carefully read and cited the reference you have provided, which is highly helpful to our work and future work. We have revised the introduction of the manuscript in line 35-37.

Line 33: There is an accidental line change that needs to be eliminated.

Response: Thank you for your valuable feedback. We have revised the format of this section of the article.

Line 34: References 2 and 3 do not seem very appropriate to me. These articles talk about trends in food, but do not have the nutritional values of fruits. They don't even talk about apples.

Response: Thank you for your valuable feedback. We have rewritten this part of the preface of the article with new references. The rewrite is at line 37-39 of the manuscript.

Figure 1: I assume "ck" is the control, but it should be spelled out in both the graphic and the text. In this case, the abbreviation saves almost no space and makes it difficult to read, even if the meaning of the abbreviation is explained the first time it is used.

Response: Thank you for your valuable feedback. In order to avoid ambiguity, we have changed the “ck” to “Control” in the whole text. The “ck” shown in the individual diagram is also illustrated in the legend.

Figure 1D; Line 74: “blueness” should be replaced by “Yellowness”, like in the graph.

Response: Thank you for your valuable feedback. We have modified the legend and pictures in Figure 1 to avoid ambiguity.

Lines 83-84: The hue angle does not indicate the amount of accumulated pigments, but the type of pigments. This parameter is a hue (color) indicator. The color intensity (chroma) could be considered an indicator of the amount of pigments. Unfortunately, this parameter (chroma) is not presented.

Response: Thank you for your valuable feedback. ​We apologize for the misrepresentation of this result, which we corrected on line 98-102 in the manuscript.

Lines 85: I assume "8 d" means "8 days". To save 3 characters, it made me spend several seconds looking for an 8d figure, until I understood the meaning of the abbreviation. If authors want readers to read their article easily and with pleasure they should avoid excessive use of abbreviations.

Response: Thank you for your valuable feedback. ​We apologize for the confusion caused by these abbreviations and have changed the manuscript of the article to replace “8d” with “8 days”.

  1. Materials and Methods: The authors do not adequately describe the process the fruits went through, nor the exact way in which cell samples were taken. Cells were taken from how many fruits? Were the fruits from which the cells were removed discarded, or did they follow the procedures after being “mutilated”?

Response: Thank you for your valuable feedback. We modify the Materials and Methods section of the Fruit material and experimental treatment to describe the number of experimental repetitions and the content sampled. The revision is in line 439-443 of the article.

Lines 508-509: I think the conclusion is too affirmative, even with regard to the apple tree. It is still much more exaggerated when trying to conclude that the treatment can be applied to all horticultural crops.

Response: Thank you for your valuable feedback. We have revised the conclusion of the paper, the content of which is in line 565-568 of the manuscript.

We genuinely value your constructive feedback, and we are committed to making the necessary revisions to improve the manuscript significantly. We are confident that the revised version will meet the journal's standards and contribute meaningfully to the field of apple coloration research. Thank you once again for your time and guidance. We look forward to resubmitting the revised manuscript for your consideration.

Round 2

Reviewer 3 Report

The article has been improved, but it still has some problems.

None of the substances used was used as a colorant, although one of them was used as a colorant under other circumstances. Colorants act on any substance, including inert ones such as plastic or glass. Dyes/colorants do not affect gene expression. They simply “paint” the object/material. Therefore, I think you should eliminate the word "colorant" throughout the article.

English is not my mother tongue. From what I can judge, the authors should make some improvements to the English of the article

Author Response

Point 1: The article has been improved, but it still has some problems.

None of the substances used was used as a colorant, although one of them was used as a colorant under other circumstances. Colorants act on any substance, including inert ones such as plastic or glass. Dyes/colorants do not affect gene expression. They simply “paint” the object/material. Therefore, I think you should eliminate the word "colorant" throughout the article.

Response: ​Thank you for taking time out of your busy schedule to review this manuscript, and we have noted the attached information about the modified version you uploaded. We are grateful to you for your careful help in correcting this manuscript. Revised portion are marked in red underline in the manuscript.

According to your suggestions, we have been make the following modifications

  • We have removed the word "colorant" from the manuscript and replaced it with the appropriate phrase in the corresponding place.
  • The highlighted portions that were previously indicated have also been revised to guarantee accurate and clear descriptions throughout the manuscript.
